# Cardiometabolic Associations between Physical Activity, Adiposity, and Lipoprotein Subclasses in Prepubertal Norwegian Children

**DOI:** 10.3390/nu13062095

**Published:** 2021-06-19

**Authors:** Tarja Rajalahti, Eivind Aadland, Geir Kåre Resaland, Sigmund Alfred Anderssen, Olav Martin Kvalheim

**Affiliations:** 1Department of Chemistry, University of Bergen, N-5020 Bergen, Norway; Tarja.Kvalheim@RKHR.NO; 2Førde Health Trust, N-6807 Førde, Norway; 3Red Cross Haugland Rehabilitation Centre, N-6968 Flekke, Norway; 4Department of Sport, Food and Natural Sciences, Western Norway University of Applied Sciences, N-6851 Sogndal, Norway; eivind.aadland@hvl.no (E.A.); Geir.Kare.Resaland@hvl.no (G.K.R.); s.a.anderssen@nih.no (S.A.A.); 5Center for Physical Active Learning, Faculty of Education, Arts and Sports, Western Norway University of Applied Sciences, N-6851 Sogndal, Norway; 6Department of Sports Medicine, Norwegian School of Sport Sciences, N-0806 Oslo, Norway

**Keywords:** cardiometabolic health, physical activity, obesity, children, principal component analysis, multivariate pattern analysis, linearly dependent covariates

## Abstract

Lipoprotein subclasses possess crucial cardiometabolic information. Due to strong multicollinearity among variables, little is known about the strength of influence of physical activity (PA) and adiposity upon this cardiometabolic pattern. Using a novel approach to adjust for covariates, we aimed at determining the “net” patterns and strength for PA and adiposity to the lipoprotein profile. Principal component and multivariate pattern analysis were used for the analysis of 841 prepubertal children characterized by 26 lipoprotein features determined by proton nuclear magnetic resonance spectroscopy, a high-resolution PA descriptor derived from accelerometry, and three adiposity measures: body mass index, waist circumference to height, and skinfold thickness. Our approach focuses on revealing and validating the underlying predictive association patterns in the metabolic, anthropologic, and PA data to acknowledge the inherent multicollinear nature of such data. PA associates to a favorable cardiometabolic pattern of increased high-density lipoproteins (HDL), very large and large HDL particles, and large size of HDL particles, and decreasedtriglyceride, chylomicrons, very low-density lipoproteins (VLDL), and their subclasses, and to low size of VLDL particles. Although weakened in strength, this pattern resists adjustment for adiposity. Adiposity is inversely associated to this pattern and exhibits unfavorable associations to low-density lipoprotein (LDL) features, including atherogenic small and very small LDL particles. The observed associations are still strong after adjustment for PA. Thus, lipoproteins explain 26.0% in adiposity after adjustment for PA compared to 2.3% in PA after adjustment for adiposity.

## 1. Introduction

Lipoprotein subclass patterns, obesity, and physical activity are associated to metabolic health. Lipoproteins can be quantified at subclass levels by high-performance liquid chromatography (HPLC) [1] or nuclear magnetic resonance spectroscopy [2]. The association of the subclass lipoprotein profile to cardiovascular health in adults is well established [3,4,5,6]. A healthy profile is characterized by high concentrations of high-density lipoproteins (HDL) and large HDL particles, and low concentrations of low-density lipoproteins (LDL), small LDL particles, very-low-density lipoproteins (VLDL), and large VLDL particles, resulting in large average size of HDL and LDL particles and a low average size of VLDL particles. Small LDL particles have for a long time been recognized as atherogenic [3,7]. Physical activity (PA) and exercise correlate positively to the healthy lipoprotein pattern [8,9,10,11,12], while overweight and obesity, as reflected in various adiposity measures, display an opposite association pattern [1,13,14,15,16,17].

The similar but inverse association pattern of adiposity and PA to the healthy lipoprotein pattern, and the association between PA and adiposity, makes it challenging to quantify the relative strength of association of PA and adiposity to the lipoprotein subclass pattern. As pointed out by Kelly et al. [12], resolving this issue represents an important research challenge. Net association patterns are needed to estimate the independent association of PA and adiposity to the lipoproteins. Adjustment for adiposity is often made by relying on a single measure, for instance, body mass index (BMI), but this option is not available for high-resolution multicollinear PA descriptors obtained from accelerometry [18,19]. A reduction to just one or a few intensity regions to reduce the multicollinearity problem leads to loss of information by not capturing the complexity of the underlying structure in the PA data. Thus, methods to handle covariates and derive net association patterns between the variables of interest are needed to adjust for such multicollinear descriptors. We use a modification of a projection algorithm [20] to accomplish this task. By decomposing multicollinear covariates into orthogonal linear combinations and use these for adjustment, we obtain “net” association patterns even for covariates possessing linear dependency. 

Our primary aim with the present paper is to reveal “net” predictive subclass lipoprotein association patterns to adiposity and PA in children and quantify the relative strength of association of the lipoproteins to adiposity and PA. We examine these associations in children since, despite the growing body of evidence for the connection between these factors early in life to cardiometabolic health later in life [21,22], few studies are available for associations of PA and adiposity to lipoprotein subclass patterns in children. Bell et al. [23] and Jones et al. [24] studied the association of subclass lipoproteins to simple PA descriptors for adolescents and children, respectively. Okuma et al. [25] studied associations between abdominal adiposity and HDL subclasses, while we previously investigated the associations of lipoproteins and BMI to aerobic fitness in a small cohort of children [26]. However, no results are available for the joint and independent associations of PA and adiposity to the important cardiometabolic pattern of lipoprotein subclasses in children.

## 2. Materials and Methods

### 2.1. Study and Participants

We used baseline data from the Active Smarter Kids (ASK) study [27], including 1129 5th graders (94% of those invited) from 57 schools in Western Norway. Of these, 841 children provided valid data on all relevant variables and were used in the present study.

Our procedures and methods conform to ethical guidelines defined by the World Medical Association’s Declaration of Helsinki and its subsequent revisions. The South-East Regional Committee for Medical Research Ethics in Norway approved the study protocol. We obtained written informed consent from each child’s parents or legal guardian and the responsible school authorities prior to testing. The study is registered in Clinicaltrials.gov with identification number: NCT02132494.

### 2.2. Lipoprotein Subclasses

Overnight fastening serum samples were obtained and stored at −80 °C according to a standardized protocol [28] and shipped on dry ice for laboratory analyses. Serum lipoprotein profiles were characterized by 26 measures: Concentrations of total cholesterol (TC), triglyceride (TG), chylomicrons (CM), very-low-density lipoproteins (VLDL), low-density lipoproteins (LDL), high-density lipoproteins (HDL), two subclasses of CM (CM-1 and CM-2), five subclasses of VLDL (VLDL-L1, VLDL-L2, VLDL-L3, VLDL-M, VLDL-S), four subclasses of LDL (LDL-L, LDL-M, LDL-S, LDL-VS), six subclasses of HDL (HDL-VL1, HDL-VL2, HDL-L, HDL-M, HDL-S, and HDL-VS), and the average particle size of VLDL, LDL, and HDL subclasses. The subclasses are labeled according to the classification of Okazaki et al. [1], except that we have merged their three subclasses of very small LDL particles and their two subclasses of very small HDL particles. Following the terminology of Ozaki et al., the abbreviations VL, L, M, S, and VS imply very large, large, medium, small, and very small particles. Some of the VL subclasses were divided further into subclasses in accordance with the classification by Okazaki et al. We calculated triglyceride and cholesterol separately and independently for all subclasses using the approach described below but finally combined them into one subclass representing the total concentration of each subclass.

The 26 lipoprotein particle measures were predicted from partial least squares (PLS) regression [29] models obtained by calibrating proton nuclear magnetic resonance (NMR) spectra to results obtained from high-performance liquid chromatography (HPLC). One hundred six serum samples were used in the calibration. Monte Carlo repeated resampling was used to validate the models with respect to predictive performance [30]. The HPLC analyses of the 106 calibration samples were performed by Skylight Biotech (Akita, Japan) according to the procedure by Okazaki et al. [1]. Proton NMR of all 841 samples was performed at the Magnetic resonance core facility (NTNU, Trondheim) by a standard procedure [31] using a Bruker Avance III 600 MHz spectrometer, equipped with a QCI CryoProbe and an automated sample changer (SampleJet) (Bruker BioSpin GmbH, Karlsruhe, Germany). Details of sample preparation, NMR conditions, and processing of spectra can be found in Jones et al. [24].

### 2.3. Physical Activity

PA was measured using the ActiGraph GT3X+ accelerometer (Pensacola, FL, USA) [32] worn at the waist over seven consecutive days, except during water activities (swimming, showering) or while sleeping. We developed a PA descriptor of time (minute/day) spent in 23 intensity intervals from the vertical axis covering the PA intensity spectrum as shown previously [19]). The intervals used for the descriptor were 0–99, 100–249, 250–499, 500–999, 1000–1499, 1500–1999, 2000–2499, 2500–2999, 3000–3499, 3500–3999, 4000–4499, 4500–4999, 5000–5499, 5500–5999, 6000–6499, 6500–6999, 7000–7499, 7500–7999, 8000–8499, 8500–8999, 9000–9499, 9500–9999 and ≥10,000 counts per minute (cpm).

### 2.4. Adiposity

We used three measures of adiposity. BMI (kg/m2) was calculated as body mass divided by the squared height. Body mass was measured (when children were in light clothing) to the nearest 0.1 kg with an electronic scale (Seca 899, SECA GmbH, Hamburg, Germany). Height was measured (when children were in their stockinged feet) to the nearest 1 mm with a transportable stadiometer (Seca 217, SECA GmbH, Hamburg, Germany). Waist circumference (WC) was measured twice between the lowest rib and the iliac crest to the nearest 0.5 cm with the child’s abdomen relaxed at the end of a gentle expiration using an ergonomic measuring tape (Seca 201, SECA GmbH, Hamburg, Germany). If the difference between measurements was >1 cm, a third measurement was taken. The average of the two closest measurements was used for analyses. We calculated waist to height ratio (WC/H), which was used in the analyses. Skinfold thickness was measured at the left side of the body using a Harpenden skinfold caliper (Bull: British Indicators Ltd., West Sussex, UK). Two measurements were taken at each position (biceps, triceps, subscapular, and suprailiac). If the difference between measurements was >2 mm, a third measurement was obtained. The total sum of the average of the two closest measurements for each site was used for analysis.

### 2.5. Data Analysis

Our data-analytical approach consists of 4 steps:

Step 1—Pretreatment of data. It is not a necessary assumption that the variables are normally distributed, but the Monte-Carlo resampling method used to determine the number of PCA or PLS components with predictive information produces more stable models if the variables are approximately normally distributed. The raw data for all variables were therefore log-transformed, mean-centered, and standardized to unit variance prior to adjustments, but with no further pretreatment before the multivariate analyses. After log transformation, normal probability plots showed that only CM, VLDL, a few of their subclasses, and TG still deviated from a normal distribution.

Step 2—Adjustment for covariates. We used the projection method [20] to adjust for all variables jointly to allow for the determination of net association patterns in step 3 and 4. Models were calculated for data adjusted only for age and sex and for models further adjusted for either adiposity or PA. Age and sex had almost zero correlation and weak correlations to the other variables and could therefore be adjusted directly by variable projection [20]. However, for the strongly multicollinear PA and adiposity descriptors, we used principal component (PC) scores [33] for adjustment. The number of components was estimated using Monte-Carlo resampling with 100 repetitions, each time randomly leaving out 25% of the data and predicting these left-out values for an increasing number of PCs. The number of PCs producing the lowest total deviation between measured and predicted values over the 100 repetitions was chosen to represent the multicollinear descriptors for the adjustment of data in step 3 and 4. By this approach, the PA descriptor of 23 variables was reduced to 4 PCs explaining 63.3, 16.5, 8.6, and 4.5% of the total variance in PA, and the adiposity descriptor of 3 variables was reduced to a single PC explaining 88.1% of the total variance in the adiposity variables. The score vectors for these PCs, which predictively describe PA and adiposity, were subsequently used for adjustment by the projection method [20].

Step 3—Exploratory analysis by principal component analysis (PCA). PCA is a recognized method to reveal and visualize correlation patterns in multicollinear data without having to assume any prior hypotheses about the data [33]. PCA maximizes the variance under the constraint of mutual orthogonality between PCs. Thus, the first PC explains most of the variance in the data, and less variance is explained by the following PCs. Each PC is a linear combination of all variables, and the coefficients (loadings) display the covariances between the variables quantitatively. If variables are standardized to unit variance, the loadings correspond to partial correlations. We used loading plots to display association patterns of variables. 

Step 4—Multivariate pattern analysis. To further examine the net association patterns revealed by PCA, we used multivariate pattern analysis [18,19] for regression modeling with lipoproteins as explanatory variables and either adiposity or PA as outcome represented by the PCs with predictive information obtained in step 2. Multivariate pattern analysis proceeds as follows: To handle the strong multicollinearity in lipoprotein variables, we used PLS regression [29]. The number of PLS components was determined by a significance test based on 1000 models calculated by repeated Monte-Carlo resampling [30]. Post-processing of the PLS models with target projection (TP) [34,35] provided a single predictive vector for the lipoproteins quantitating the associations to the predicted outcome. For model interpretation, we calculated selectivity ratios (SRs) as the ratio of explained variance on the target component to the total variance of the target model. This procedure differs from our earlier procedure [35,36], where we used the residual variance in the denominator. By relating SR to the total variance, we obtain a direct measure of explained predictive variance in the explanatory variables with the same ability for visualization and interpretation as for the original SR plot.

## 3. Results

### 3.1. Descriptive Statistics for the Variables

The 841 children (50% boys) were (mean ± standard deviation) 10.2 ± 0.3 years old, had a BMI of 18.0 ± 3.0 kg/m^2^, a WC/H of 0.43 ± 0.05, and a skinfold thickness of 49.8 ± 26.4 mm. Mean and standard deviation for concentrations of lipoprotein classes were (mmol/L): TC (4.35 ± 0.66), TG (0.74 ± 0.38), CM (0.036 ± 0.051), VLDL (1.07 ± 0.51), LDL (2.42 ± 0.50), HDL (1.57 ± 0.25), CM-1 (0.021 ± 0.031), CM-2 (0.014 ± 0.020), VLDL-L1 (0.059 ± 0.064), VLDL-L2 (0.14 ± 0.14), VLDL-L3 (0.36 ± 0.18), VLDL-M (0.26 ± 0.11), VLDL-S (0.25 ± 0.06), LDL-L (0.79 ± 0.17), LDL-M (1.04 ± 0.22), LDL-S 0.42(± 0.09), LDL-VS (0.18 ± 0.04), HDL-VL1 (0.020 ± 0.004), HDL-VL2 (0.067 ± 0.031), HDL-L (0.36 ± 0.16), HDL-M (0.56 ± 0.08), HDL-S (0.40 ± 0.04), and HDL-VS (0.16 ± 0.01). Average particle size (nm) for the main lipoprotein classes was: VLDL (42.5 ± 2.8), LDL (25.77 ± 0.13), and HDL (10.87 ± 0.21). Details of the characteristics of the 23 PA variables used in this study can be found in Aadland et al. [19].

### 3.2. PCA of Lipoproteins Adjusted for Age and Sex

Figure 1 shows the variable loadings on the first PC after adjusting adiposity, lipoprotein, and PA variables for age and sex.

The loading plot reveals increasingly strong positive correlations between the intensity regions of the PA descriptor up to 6000–7000 cpm where correlation levels off and slowly declines. The average size of LDL and HDL particles and concentration of HDL, and very large-, large-, and medium-size HDL particles correlate positively to this PA pattern. All the other lipoprotein and all adiposity measures correlate negatively to this PA pattern and positively to sedentary time.

### 3.3. PCA of Lipoproteins and PA Adjusted for Age, Sex and Adiposity

Adjustment for the covariates age, sex, and adiposity removed 9.4% of the original variance in both the lipoprotein and PA variables. Figure 2 shows the variable loading pattern on the first PC after adjustment.

We observe a strong partial correlation between almost all PA variables with a flat maximum around the intensity level 5500–6000 cpm. This PA correlation pattern differs from the pattern in the data not adjusted for adiposity (Figure 1), where we noticed a pronounced increase in associations between PA variables up to 6500–7000 cpm. The observation complies with the variance plot (Appendix A, Figure A1); that adiposity correlates strongest with high-intensity PA, and that adjustment for adiposity consequently weakens this association. Associations between PA and lipoproteins also weaken after adjusting for adiposity, but the overall association pattern persists: The triglyceride-rich lipoproteins, CM, VLDL, and their large subclasses correlate negatively to PA, while HDL concentration, average particle size of HDL and LDL, and large and very large subclasses of HDL correlate positively to PA. Thus, the correlation pattern among the PA and lipoprotein variables appears robust to the removal of the variation associating with adiposity, but the strength of correlations is considerably weakened.

### 3.4. PCA of Lipoproteins and Adiposity Adjusted for Age, Sex and PA

In accordance with the visual observations from the variance plot (Appendix B, Figure A2), adjustment for the covariates age, sex, and PA removed considerably more of the variance in adiposity than in the lipoprotein variables, i.e., 20.4% for adiposity compared to only 5.0% for the lipoprotein features.

A bivariate loading plot was used for the interpretation of this analysis since it has better interpretability when variables have high partial correlations on more than one PC. Figure 3 shows such a loading plot visualizing the net association patterns of lipoproteins with adiposity variables on PC1 and PC3. These two PCs jointly explained 50.9% of the net variance. PC2 displayed almost no association to adiposity, only mutual associations between lipoproteins, and was therefore not further examined.

The bivariate loading plot (Figure 3) reveals a strong inverse association between adiposity and the average size of HDL and LDL particles as well as the total concentration of HDL and concentrations of very large-, large-, and medium-sized HDL particles. The group (c) of lipoprotein subclass features locates near adiposity in the loading plot, and thus positively correlates to adiposity, embraces concentrations of LDL, small and very small particles of HDL and LDL, and medium-sized LDL particles. The triglyceride-rich CM and VLDL particles, group (b), associate weakly with adiposity.

### 3.5. Multivariate Pattern Analysis of Lipoproteins

The PCs used to adjust for PA and adiposity were used as outcome variables in multivariate pattern analysis. Figure 4 displays the loadings on PC1 for the PA variables, explaining almost two-thirds of the total variance in PA.

We observe (Figure 4) a pattern of positive correlations increasing to a maximum at 6000–6500 cpm, except for sedentary time (Min_0), which correlates negatively to the other PA variables. Adjusted for age, sex, and adiposity, multivariate pattern analysis with PC1 extracted for PA as outcome and the lipoproteins as explanatory variables provided a model explaining 2.3% of the net variance in PC1. The SR plot (Figure 5) for this model shows that the net PA pattern correlates negatively to all the triglyceride-rich lipoproteins (CM, VLDL, and all VLDL subclasses except VLDL-S) and positively to the concentration of HDL, very large-, large-, and medium-sized HDL particles and average HDL particle size. None of the LDL features associate to the net PA signal. Neither PC2 nor PC3 extracted for PA carried information associating predictively to the lipoproteins. They displayed mutual associations between PA variables.

PC1 for adiposity accounted for 88.1% of the total variance of BMI, WC/H, and skinfold and was used as outcome with the lipoproteins as explanatory variables. This PC (not shown) displayed a pattern of positively correlated loadings of almost equal size for the three adiposity measures. After adjustment for age, sex, and PA, the lipoproteins explained 26.0% of the variance in adiposity. The SR plot (Figure 6) shows the inverse association pattern for adiposity to CM, VLDL, and HDL lipoproteins to the pattern observed for PA. In addition, we notice a positive correlation of small and very small atherogenic LDL particles to adiposity.

### 3.6. Summary of Findings

By using a novel approach that allowed adjustment for linear dependent covariates, we determined the net association pattern of PA and adiposity to lipoprotein subclasses in children. Adiposity and PA associated almost inversely to a healthy cardiometabolic lipoprotein subclass profile of CM, VLDL, and HDL subclasses. In addition, adiposity associated to the atherogenic small- and very-small LDL particles [3,4,5,7]. The strength of the net association of adiposity and PA to the lipoprotein pattern indicated a detrimental association of adiposity to cardiovascular health that dominated over the positive association of PA to cardiovascular health.

## 4. Discussion

The association pattern of lipoproteins to PA and adiposity (Figure 1) are mainly consistent with associations between the lipoprotein profile, aerobic fitness, and BMI previously observed in a smaller cohort of Norwegian children in the same age group [26]. The net association pattern between lipoproteins and PA (Figure 2) also mostly complies with previous investigations where corresponding features have been examined [8,9,10,23,24,25]: Positive associations of PA to HDL, very large and large HDL particles, and the average size of HDL particles, and negative association to small and very small HDL particles. This HDL subclass pattern associates positively with cardiovascular health [25]. Furthermore, we found a negative association between the concentration of TG, VLDL, and large VLDL particles and the average size of VLDL particles which is also consistent with previous studies [8,9,10,23]. However, for intervention studies on adults, both Kraus et al. [8] and Halverstadt et al. [9] observed a strong effect of PA on the concentration of LDL and the atherogenic small LDL particles, as well as on the average LDL particle size. The results were obtained with minimal weight loss [8] and independent of change in body fat [9], respectively. Although we observed the same pattern, the net associations to PA in children were weak (Figure 2). Reasons for this discrepancy can be differences in lipoprotein profiles between children and adults and observational versus experimental design. The adults involved in the intervention studies reporting changes in the LDL subclass pattern did regular high-amount-high-intensity exercise training [8]. This result complies with our earlier investigation of lipoprotein associations to PA for a population of healthy adults [37]. Though not statistically significant due to the small sample size, moderate and vigorous PA associated to a healthy LDL pattern of less small LDL particles and reduced concentrations of LDL [37].

The association pattern of lipoproteins to PA obtained by using PC1 for PA as outcome in regression analysis (Figure 5) confirmed the findings from our exploratory analysis using PCA. The SR plot displayed an association pattern that is very similar to the pattern previously observed for the Andersen aerobic fitness test in a smaller cohort of children [26]. The absence of a significant association pattern between PA and LDL lipoprotein features was also evident. Note also that the PA loading pattern on PC1 (Figure 4), which was used as outcome and thus associated to the pattern displayed in Figure 5, implied a positive association of PA to cardiovascular health over the whole intensity scale, except sedentary time. Associations peaked at 6000–6500 cpm and slowly declined with increasing intensity. Since the time spent in low-intensity PA was three times higher than time spent in moderate and high-intensity PA [19] for our cohort, the PA pattern in Figure 4 highlights the importance of moderate and high-intensity PA for the cardiovascular healthy lipoprotein association patterns in Figure 5.

The net association pattern for adiposity to lipoproteins revealed a very different picture from the association pattern obtained for PA. The variable loading plot of PC3 vs. PC1 (Figure 3) showed that the atherogenic LDL subclass pattern correlated positively to adiposity. In contrast, the healthy cardiovascular pattern for CM, VLDL, and HDL subclasses obtained for PA correlated negatively to adiposity. This became even more evident from the SR plot (Figure 6) obtained with PC1 combining the three adiposity measures into a single composite adiposity outcome. In particular, the association of adiposity was strong for the small LDL particles. The negative associations of adiposity to the HDL features associating to cardiovascular health was also evident. This is in line with the findings of Slyper at al. [6] for a cohort of 61 obese nondiabetic adolescents. They found significant correlations of thickening of the intima-media of the carotid artery in the pediatric population to BMI z-score and concentrations of LDL, VLDL, HDL, and subclasses of HDL with an association pattern matching the one we found in our investigation. The inverse relationship between adiposity and concentration of HDL also agrees with an investigation of 917 non-diabetic Cherokee Indian children and adolescents carried out by Blackett et al. [38]. These associations were already present in 5–9 years old children (both boys and girls) and persisted in older children (10–14 years old) and adolescents (15–19 years old). Spinneker et al. [39] observed the same inverse relationship between HDL and both BMI and body fat for a European cohort, including 1076 adolescents (12–18 years old). They also observed an increase in LDL concentrations with increasing BMI and body fat. Kaitosaari et al. [40] divided a cohort of 176 healthy 7-year-old children into two groups below and above the median of the average LDL particle size. They found no significant association between adiposity and the average size of the LDL particles, but a positive association of concentration of HDL to average size of LDL particles and concluded that the atherogenic LDL pattern of high concentration of small LDL particles develops after puberty. This result contradicts the findings of Jones et al. [24] of a significant negative association between adiposity and the average size of LDL particles in prepubertal children. However, the small sample size and the group-wise comparison used in the statistical analysis by Kaitosaari et al. [40] may be responsible for the discrepancy.

Most of the lipoprotein subclasses associated inversely to adiposity and PA. As pointed out by Kelly et al. [12], this creates a challenging situation with respect to separating their independent associations to the lipoproteins. We have shown that this is possible without compromising on the use of high-resolution and high-informative PA descriptors, which pose problems for standard methods for data analysis. In addition, our analytical approach enables quantification of the relative strength of the net association of lipoproteins to PA and adiposity. With approximately 10 times more explained variance in adiposity compared to PA in the regression modeling of net lipoprotein association patterns, adiposity dominates over PA in the strength of association patterns to lipoprotein in the examined cohort. Since previous studies have not used methods that allow for the possibility to adjust for linear dependent covariates, we do not know if our result is generalizable to other age groups. Future studies are needed to corroborate the relative importance of PA and adiposity for a healthy lipoprotein profile in adults.

### Strengths and Limitations of Study

We included a high-resolution PA descriptor derived from objective measurements by accelerometry, three different adiposity measures, and a comprehensive lipoprotein profile for a large cohort of children. Moreover, our approach to detect and validate underlying association patterns in multicollinear data is beneficial to univariate testing of each association in isolation: Multicollinear patterns are more stable to perturbations than single associations, and we can assess whole patterns for significant predictive information instead of one association at the time. This attribute mean that we can obtain validated results for a smaller sample size with the multivariate than with a univariate approach, where we would need a larger sample size to achieve the same statistical power due to multiple testing. Our method is also able to handle linearly dependent covariates and recognizes that covariates are not error-free, which is a basic assumption when standard linear regression is used to adjust for covariates.

The cohort we analyzed consisted of a rather homogenous population of children from a geographically restricted area of Western Norway and within a narrow age range. Thus, the association patterns we found for the extensive lipoprotein profile to adiposity and PA may not be generalizable to adolescents and adults or to children from the wider genetic and environmental pool. However, for lipoprotein features where comparison is possible with previous studies, the associations to PA and adiposity generally agree across age, sex, and genetic factors [8,9,10,11,12,23,25,38,39,40].

Of the total variance in PA and adiposity, 7.2% and 11.9%, respectively, remained unused when the predictive PCs were applied for adjustment. This may potentially lead to residual confounding. To examine this possibility, we repeated all the analyses with the adjusted adiposity or PA variables incorporated. The possible impact of residual confounding was visualized by means of PCA loading plots. In all plots, the adiposity measures and the PA variables were located near the origin of the loading plots (results not shown). This observation implies that the residual variance in the adiposity and PA variables after adjustment using principal components is uncorrelated to the dominant net association patterns and will not affect model interpretation.

## 5. Conclusions

We disentangled the net association pattern of an extensive multicollinear lipoprotein profile to adiposity and PA with important cardiometabolic health implications. Our findings showed that adiposity and PA possess independent associations to the lipoprotein subclass profile. The association patterns were almost inverse but much stronger for adiposity than for PA. Our data-analytical approach to handling linear dependent covariates was crucial to achieving these results and provides a general solution to the challenge posed by metabolites associated to several strongly related factors [12]. Thus, we provide new evidence on the role of adiposity in the PA metabolomics relationship through a methodological approach that can inform future research in this field.

## Figures and Tables

**Figure 1 nutrients-13-02095-f001:**
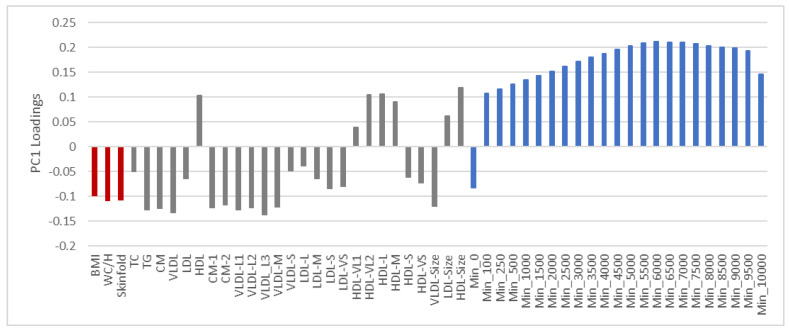
Loading plot displaying the partial correlations of lipoproteins, adiposity, and physical activity explaining 30.6% of the total remaining variance in the variables after adjusting for age and sex. The following color code is used to specify different kinds of variables: Red for adiposity, grey for lipoproteins, and blue for physical activity. For the physical activity variables, the names imply the lowest intensity levels in each intensity interval defined in Section 2.3. For instance, the name Min_0 implies the intensity interval 0–99 cpm.

**Figure 2 nutrients-13-02095-f002:**
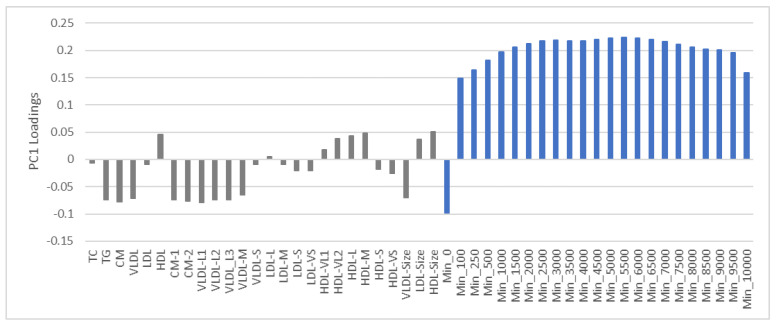
Loading plot displaying the partial correlations of lipoproteins and physical activity explaining 30.3% of the total variance in the variables after adjusting for age, sex, and adiposity. The following color code is used to specify different kinds of variables: Grey for lipoproteins and blue for physical activity. For the physical activity variables, the names imply the lowest intensity levels in each intensity interval defined in Section 2.3. For instance, the name Min_0 implies the intensity interval 0–99 cpm.

**Figure 3 nutrients-13-02095-f003:**
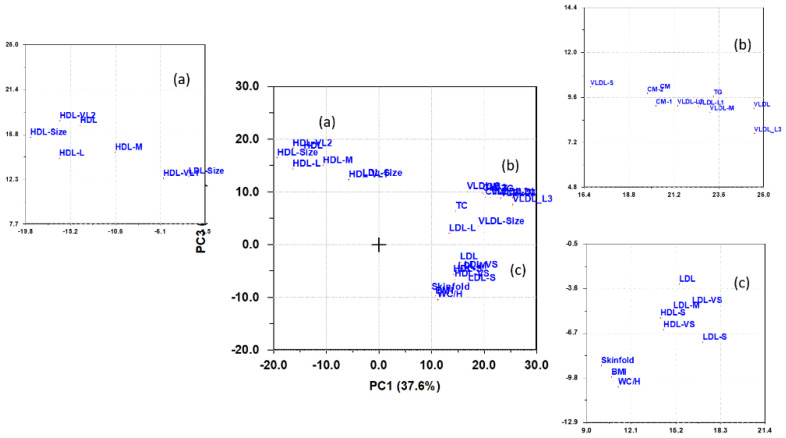
Loading plot displaying the partial correlations of lipoproteins and adiposity explaining 37.6 and 13.3% of the total variance in the variables on PC1 and PC3, respectively, after adjusting for age, sex, and physical activity. Groups of variables labeled (**a**–**c**) in the main graph are zoomed to increase resolution.

**Figure 4 nutrients-13-02095-f004:**
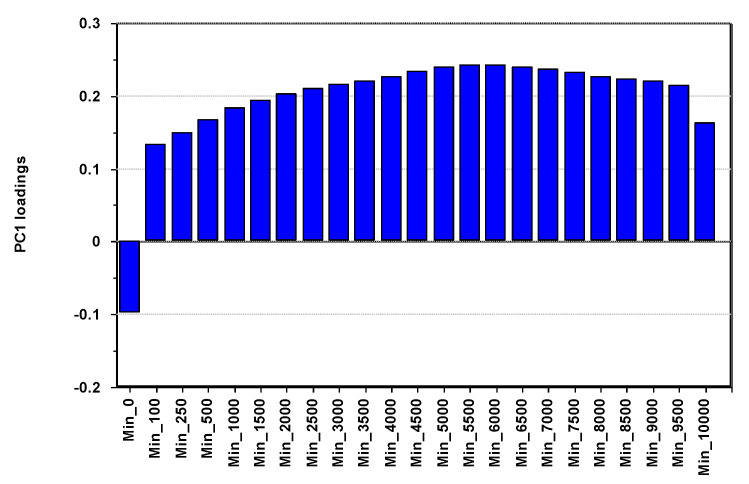
Loading plot displaying the correlation pattern of the 23 physical activity variables explaining 63.3% of the total variance in the variables without any adjustments for covariates. The variable names imply the lowest intensity levels in each intensity interval defined in Section 2.3. For instance, the name Min_0 implies the intensity interval 0–99 cpm.

**Figure 5 nutrients-13-02095-f005:**
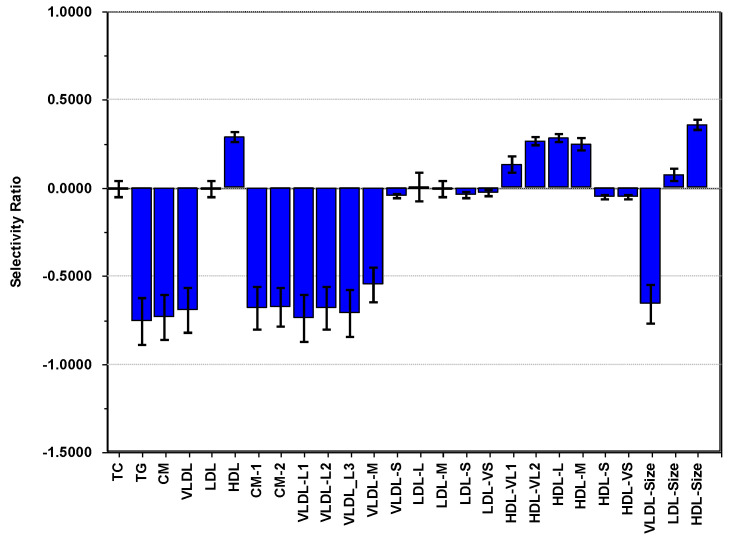
Selectivity ratio plot of a regression model with physical activity as outcome (R2Y = 0.023) and the lipoproteins, adjusted for age, sex, and adiposity, as explanatory variables.

**Figure 6 nutrients-13-02095-f006:**
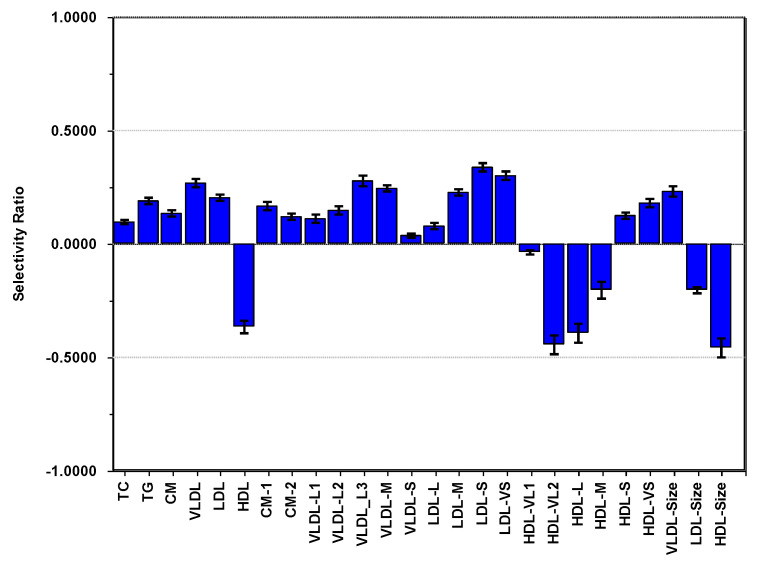
Selectivity ratio plot of a regression model with the adiposity PC1 score as outcome (R2Y = 0.260) and the lipoproteins, adjusted for age, sex, and physical activity, as explanatory variables.

## Data Availability

The datasets used in the current study are available from the corresponding author on reasonable request.

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
