# Peer review of "Cardiometabolic Associations between Physical Activity, Adiposity, and Lipoprotein Subclasses in Prepubertal Norwegian Children"

_nutrients, 2021, doi:10.3390/nu13062095_

Round 1
Reviewer 1 Report
This paper describe a very large metabolic study, which is more than adequate sample size for the purposes of this report. This paper uses modern data analytical methods to visualise complex patterns in lipoprotein profiles from Norwegian school children.
I have only a few comments.
The abstract might want to mention the analytical techniques used.
Data analysis. It might be useful to describe how data were preprocessed before PCA and PLS.
Step 3 : PCA, is a basic description of the method necessary? It has been described in thousands of papers and this article is not a special review about PCA.
Step 4 : The term multivariate pattern analysis is a bit ambiguous, most data scientists use the term multivariate pattern recognition. I found “Multivariate pattern analysis (MVPA) comprises a collection of tools that can be used to understand complex spatial disease effects across the brain.” (https://www.ncbi.nlm.nih.gov/pmc/articles/PMC5154735/) “• Broadly: Any analysis method whose inputs are patterns of activity over multiple features (i.e. voxels or points on a surface)” https://cfn.upenn.edu/stslab/wiki_links/fmri_club/mvpa/mvpa_matt.pdf. Perhaps the authors can be more specific where their definition fits in.
Figure 1, I am not sure of the colour scheme nor Figure 2 (why no red?).
Figure 3, the labels are a bit crowded.
For all the PCplots, please specify how the data were preprocessed.
Author Response
I have only a few comments.
The abstract might want to mention the analytical techniques used.
Answer: Done.
Data analysis. It might be useful to describe how data were preprocessed before PCA and PLS.
Answer: This was already mentioned as step 1 under section 2.5. However, we have clarified that after preprocessing of raw data no further preprocessing was done except adjustments. And this is mentioned in the figure caption for each plot.
Step 3 : PCA, is a basic description of the method necessary? It has been described in thousands of papers and this article is not a special review about PCA.
Answer: We have shortened the description. There are still many researchers who are not familiar with PCA.
Step 4 : The term multivariate pattern analysis is a bit ambiguous, most data scientists use the term multivariate pattern recognition. I found “Multivariate pattern analysis (MVPA) comprises a collection of tools that can be used to understand complex spatial disease effects across the brain.” (https://www.ncbi.nlm.nih.gov/pmc/articles/PMC5154735/) “• Broadly: Any analysis method whose inputs are patterns of activity over multiple features (i.e. voxels or points on a surface)” https://cfn.upenn.edu/stslab/wiki_links/fmri_club/mvpa/mvpa_matt.pdf. Perhaps the authors can be more specific where their definition fits in.
Answer: The term multivariate pattern analysis can have different meanings in different fields. Here it refers to the three-step method of i) partial least squares regression, ii) target projection, and, iii) calculation of selectivity ratios. The selectivity ratio vector is then displayed as an SR plot for model interpretation. This meaning is well established in numerous papers in the physical activity field. See, for instance, refs. 18 and 19.
Figure 1, I am not sure of the colour scheme nor Figure 2 (why no red?).
Answer: Explanation is added in figure caption. In Fig. 2, the red adiposity variables are not shown since they are adjusted for and are therefore have loadings close to zero.
Figure 3, the labels are a bit crowded.
Answer: Agree, so we have zoomed the crowded variable loading areas.
For all the PCplots, please specify how the data were preprocessed.
Answer: This was already done in section 2.5. Only adjustment was done after the initial preprocessing described in section 2.5 and adjustments are described in the figure captions.
Reviewer 2 Report
Few studies are available for association of PA and adiposity to lipoprotein subclass patterns in children. Authors employed a novel approach to adjust for covariates, for detecting the “net “ patterns and strength for PA and adiposity to the lipoprotein profile.
They obtained the results, which were clearly shown that were available for the joint and independent associations of PA and adiposity to the cardiometabolic important pattern of lipoprotein subclasses in children. Their present studies may contribute to the role of adiposity in the PA metabolomics relationship, as they mentioned.
Author Response
It appears that the reviewer was happy with our work.